# Effects of Tongue Pressure on Cerebral Blood Volume Dynamics: A Functional Near-Infrared Spectroscopy Study

**DOI:** 10.3390/brainsci12020296

**Published:** 2022-02-21

**Authors:** Hidemasa Miyata, Ryouji Tani, Shigeaki Toratani, Tetsuji Okamoto

**Affiliations:** 1Department of Molecular Oral Medicine and Maxillofacial Surgery, Graduate School of Biomedical & Health Sciences, Hiroshima University, Kasumi 1-2-3, Minami-ku, Hiroshima 734-8553, Japan; smile737voice@yahoo.co.jp (H.M.); tora@hiroshima-u.ac.jp (S.T.); 2Oral and Maxillofacial Surgery, Hiroshima University Hospital, Kasumi 1-2-3, Minami-ku, Hiroshima 734-8553, Japan; 3School of Medical Sciences, University of East Asia, 2-1 Ichinomiyagakuenchō, Shimonoseki 751-8503, Japan; tetsuok@hiroshima-u.ac.jp

**Keywords:** tongue pressure measurement, near-infrared spectroscopy, cerebral cortex hemodynamics, cerebral blood volume

## Abstract

Tongue pressure measurement (TPM) is an indicator of oral function. However, the association between tongue pressure and cerebral activation remains unclear. We used near-infrared spectroscopy (NIRS) to examine the correlation between cerebral cortex activation and tongue pressure stimulation against the anterior palatal mucosa. We measured voluntary maximum tongue pressure (MTP) using a TPM device; a pressure value of approximately 60% of the MTP was used for the experimental tongue pressure (MTP_60%_). We examined the effect of oral functional tongue pressure stimulation against the anterior palatal mucosa on cerebral activation using NIRS in 13 adults. Tongue pressure stimulation caused significant changes in cerebral blood flow in some areas compared with controls (*p* < 0.05). We performed a correlation analysis (*p* < 0.05) between MTP_60%_ and changes in oxygenated hemoglobin in all 47 NIRS channels. MTP_60%_ triggered activation of the right somatosensory motor area and right dorsolateral prefrontal cortex and deactivation of the anterior prefrontal cortex (APFC). TPM balloon-probe insertion in the oral cavity activated the bilateral somatosensory motor area and deactivated the wide area of the APFC. Moreover, MTP_60%_ via the TPM balloon probe activated the bilateral somatosensory and motor cortex areas. Tongue pressure stimulation changes cerebral blood flow, and NIRS is useful in investigating the relationship between oral stimulation and brain function.

## 1. Introduction

In ageing societies, cognitive decline and physical deterioration pose serious health-related challenges and are symptomatic of various diseases. Among physical challenges, oral functional deterioration is a cause of mortality owing to its relationship with both dysphagia [1] and respiratory disorders. Oral functional deterioration assessments have been developed, such as tongue pressure measurement (TPM), which is used as a simple, objective indicator in Japan. Previous studies have utilized TPM for detecting swallowing disorders caused by low tongue pressure [2] and for oral functional training (OFT) in dysphagic patients [1]. Furthermore, TPM can predict pneumonia in acute stroke patients [3] and is effective in oral functional assessment and OFT. Conversely, it was suggested that mastication increases regional cerebral blood flow (CBF) [4]. In addition, sensory stimulation of the oral mucosa increases regional CBF [5]. Therefore, OFT may improve cognitive function [6,7]. However, the mechanisms by which oral functional stimulation affects regional cerebral function remain unknown [8,9].

Noninvasive functional neuroimaging techniques have been developed to assess human brain function. Cerebral functional activity is detected using surrogate biomarkers such as neural electromagnetic activities, CBF, and oxygenation. In particular, CBF- and oxygenation-based neuroimaging techniques have been developed to elucidate cerebral function based on neurovascular coupling [10].

Fox et al. [11] first reported the use of the positron emission tomography (PET) technique for this purpose. Furthermore, researchers [12] devised the blood oxygenation level dependent (BOLD) contrast method, which entails changing the signals in functional magnetic resonance imaging (fMRI) without the use of contrast agents. fMRI provides a clear, three-dimensional depiction of cerebral activity with sufficient spatial resolution. Hence, fMRI has been used for the noninvasive identification of cerebral function of the sensory [13] and motor areas [14]. Several neurofunctional studies have been conducted to date; however, fMRI studies need a strong magnetic field, the restriction of head movement, and supine positioning of the participant. Therefore, it is difficult to measure cerebral function under daily physiological conditions [15].

In contrast, near-infrared spectroscopy (NIRS) is based on the biopermeability of near-infrared light. NIRS can detect the hemodynamics of the cerebral cortex caused by neurovascular coupling. In 1977, Jöbsis [16] first noninvasively measured CBF and tissue oxygenation in beagles and humans. After its development in 1993, NIRS has been used to measure cerebral functional hemodynamics [17], and it is currently used in neurosurgery and for assessing both domain-specific, language-related abilities [18] and depression [19]. Moreover, Japanese health insurance includes a provision for the use of NIRS for the differential diagnosis of depression [20]. NIRS can predict and evaluate drug efficacy in children with attention-deficit hyperactivity disorder [21]. Taken together, NIRS has become a new neuroimaging modality for the noninvasive detection of the hemodynamics of the cerebral cortex under routine physiological conditions.

The purpose of this study was to examine regional CBF induced by tongue pressure against the anterior palatal mucosa and OFT with TPM stimulation. We used NIRS due to its greater flexibility in task assignments compared to other functional neuroimaging techniques and evaluated the potential of NIRS as an on-site, objective indicator of the rehabilitation effect of tongue pressure OFT.

## 2. Materials and Methods

### 2.1. Participants

Thirteen healthy adult volunteers (men, 10; women, 3; age (mean ± SD), 29.3 ± 3.6 years) were recruited. The participants had no unstable illness, were not taking any medication, had not consumed alcohol, and provided written informed consent after they were given information about this study. All procedures were approved by the appropriate Ethics Committee of Hiroshima University Hospital, Hiroshima, Japan, on 10 January 2017 (authorization number: E-173).

### 2.2. Maximum Tongue Pressure Measurement

The experimental stimulus, i.e., maximum tongue pressure (MTP), was evaluated using a balloon-type TPM device (TPM-02; JMS Co. Ltd., Hiroshima, Japan). The participants were seated on a chair with a backrest and instructed to hold the plastic pipe of the TPM device at the midpoint of their central incisors with their lips closed. Participants were asked to lift their tongue against the anterior palatal mucosa around the palatal rugae. The mean MTP was calculated from three measurements. The experimental tongue-pressure stimulus was determined at approximately 60% of the mean MTP (MTP_60%_), considering the effects of overshooting, artifact, fatigue, and struggling to perform the task (Table 1).

### 2.3. Practice for MTP60%

The experiment was conducted after each participant practiced reproducing the sensation of pressing the mucosa of the anterior palate by elevating the tongue with the same level of pressure felt on the palate as when the probe was pressurized with 60% MTP.

### 2.4. NIRS Measurement Items

We employed a multichannel NIRS system (ETG-7100; Hitachi Medical Corporation, Tokyo, Japan) to measure changes in cerebral hemodynamics using two wavelengths of near-infrared light (695 and 830 nm) and the accompanying 47 channels (Chs). The measurement probe had an array of 3 × 11 optical fibers, and the distance between the light-emitting unit and detector unit was 30 mm. We used an NIRS head-holder, with the emitter and detector arranged along the 3 * 5 line, and set the lower line of the NIRS head-holder along the T4-Fpz-T3 line according to the International 10–20 method of electroencephalography (EEG) electrode placement. All measurements were conducted in a silent room at the Hiroshima Innovation Center of Biomedical Engineering and Advanced Medicine (Hiroshima University, Kasumi Campus, Hiroshima, Japan).

Continuous-wave NIRS is the most used type [22], in which light absorption is detected by density changes based on the modified Beer–Lambert Law. Based on this principle, NIRS can detect hemoglobin fluctuations in the cerebral cortex 2 cm below the scalp.

We determined the arrangement between the probe and cerebral prefrontal surface using the virtual registration method [23,24]. The detector signal consisted of oxygenated hemoglobin (Oxy-Hb) and deoxygenated hemoglobin (Deoxy-Hb) data that were continuously measured. The total Hb concentration was obtained by adding the Oxy-Hb and Deoxy-Hb data.

### 2.5. Task Paradigm

The task paradigm consisted of a block design. Each block consisted of a 30 s stimulation period and a 40 s recovery period. The experimental stimulus was applied with the tongue three times for 5 s against the anterior palatal area during the 30 s stimulation period via a TPM balloon-type probe (Figure 1). The task block was repeated for four cycles. The average hemodynamic response was obtained by averaging the blocks performed four times per participant. The measurement procedure lasted for a total of 19 min.

(1)Task 1: resting tongue position for control.(2)Task 2: MTP_60%_ against the anterior palatal mucosa (5 s, three times).(3)Task 3: insertion of the TPM probe.(4)Task 4: MTP_60%_ via the TPM probe (5 s, three times).

### 2.6. Analysis of the NIRS Data

#### 2.6.1. Wave Analysis

Continuous NIRS measurement was integrated with the baseline determined using the mentioned parameters (pre-time, 10 s; recovery time, 35 s; post-time, 5 s).

#### 2.6.2. Activation Analysis

We calculated the Oxy-Hb values for all 47 Chs within the stimulation period during the task blocks (10–40 s). The mean value of the change in the Oxy-Hb concentration (from baseline) was determined using the area under the curve. The baseline for each task was the line connecting the average of the 10 s pre-time and the 5 s post-time. After Benjamini and Hochberg’s false-discovery-rate processing of the subcurve values for each of the 47 channels, multiple comparison analysis with Dunnett’s post hoc comparison was used to examine the tasks and identify the channels that significantly changed blood flow. Statistical analyses were performed using SPSS version 21 (IBM Corp., Armonk, NY, USA).

#### 2.6.3. Comparison of Sex Correlations between Hemodynamic Parameters

In order to examine whether there was a difference in the variation in CBF owing to sex, a statistical test using a paired *t*-test was performed on the subcurve values of each of the 47 channels in the men’s and women’s groups.

#### 2.6.4. Correlation Analysis

The relationship between MTP_60%_ and changes in Oxy-Hb (ΔOxy-Hb) levels in each Ch was analyzed using correlation analysis. Statistically significant Chs are depicted on the cortical surface map in Figure 2.

## 3. Results

### 3.1. Maximum Tongue Pressure

The intensity of the tongue pressure caused by MTP was evaluated using three measurements with a JMS TPM device. The results showed a mean MTP value of 41.2 kPa, which is indicative of mild oral functional stimulation. Therefore, the approximate MTP_60%_ was used in our analysis. There was no significant difference in the mean MTP between the men’s and women’s groups (Table 1).

### 3.2. Waveforms Elicited by the Tasks

Activation tendency is represented in the topography and waveforms of Ch 27 in Figure 3. Oxy-Hb was used to determine cerebral cortex activation. The classical waveform of the four experimental tasks of the participants is depicted in Figure 3. The topography shows a panoramic distribution of Oxy-Hb in the cerebral cortex.

Task 1 resulted in an unchanging Oxy-Hb condition during the control state on the topographical map (Figure 3A). Task 2 resulted in a task-related Oxy-Hb increase among the block stimulation periods (10–40 s) in the left Ch 27 (Figure 3B). The Oxy-Hb signals also increased during the insertion of the balloon-type TPM in Task 3 (Figure 3C). MTP_60%_ via the balloon-type TPM probe resulted in weaker Oxy-Hb activation than that in Task 2 in the left Ch 27 (Figure 3D).

### 3.3. Activation Analysis between Tasks

Dunnett’s post hoc test for the comparison of the stimulation and control tasks revealed significant activation in the channels concerned with oral functional stimulation. Each of the 47 Chs represented approximately one cerebral cortical region (Figure 4). Figure 4A shows significant changes in Oxy-Hb concentration that were confirmed between the tongue-press and control tasks. Significant alterations were observed in the right superior marginal gyrus (SMG), right primary sensory cortex (PSC), right primary motor cortex (PMC), and bilateral superior temporal gyri (STG). A significant increase was also observed in the right dorsolateral prefrontal cortex (DLPFC), while a significant decrease was observed in the right anterior prefrontal cortex (APFC).

As shown in Figure 4B, significant changes in Oxy-Hb were observed during the balloon-type probe-insertion and control tasks. There were significant elevations in the bilateral SMG, PSC, PMC, and STG. Wide-area deactivation of the DLPFC and APFC was also observed.

MTP_60%_ stimulation via the balloon-type-probe task resulted in bilateral Ch activations that differed significantly from those observed during the control task. Significant elevations were also observed in the left PSC, bilateral SMG, PMC, and right DLPFC (Figure 4C).

### 3.4. Comparison of Sex Correlations between Hemodynamic Parameters

We used a corresponding *t*-test to analyze sex and MTP but found no significant difference. We also examined cerebral blood flow in the men’s and women’s groups using a corresponding *t*-test, and Ch 24 (*p* = 0.008) in Task 3 and Ch 42 (*p* = 0.043) in Task 4 showed a significant difference.

### 3.5. Correlation Analysis

The correlation analysis between MTP_60%_ and ΔOxy-Hb revealed significant channels: Ch 26 (*r* = 0.636, *p* = 0.049) and Ch 39 (*r* = 0.645, *p* = 0.001). The correlation analysis between MTP_60%_ via the balloon-type probe of TPM and ΔOxy-Hb revealed significant channels: Ch 45 (*r* = 0.555, *p* = 0.049) and Ch 39 (*r* = 0.791, *p* = 0.017) (Figure 5).

## 4. Discussion

This study showed that tongue pressure against the anterior palatal mucosa and TPM have various effects on regional CBF. MTP_60%_ against the anterior palatal mucosa was predominantly localized to the right temporal cortex. In a previous study, oral stimulation of the first transverse palatine ridge resulted in bilateral activation of the primary somatosensory cortex [25]. However, our study suggests that bilateral activations of the somatosensory and motor cortices are attenuated by the adaptation of physiological conditions. In addition, hearing the researcher’s cues and performing voluntary isometric lingual exercises may simulate verbal–lingual coordination [26,27,28].

The participants were examined for blood flow fluctuations in the cerebral cortex when tongue pressure was applied, and these data may serve as a basis for excluding the effects of unconscious tongue pressure in an fNIRS study.

The TPM balloon-type probe activated the bilateral temporal lobes and deactivated the wide area of the PFC, which is in line with previous findings [5]. MTP_60%_ via the TPM balloon-type probe activated the bilateral primary sensory and motor cortices, which is consistent with oral ball-rolling tasks [29]. This study suggests that each oral function affects CBF.

A correlation analysis revealed a significant, positive relationship between the MTP_60%_ and ΔOxy-Hg values at the channels over the left DLPFC and right MTG. The DLPFC plays an important role in emotional and motivational tendencies (i.e., vitality [30]), and this specific region may be a biomarker for tongue function.

Since CBF responses may be altered by ageing, we examined changes in CBF in the cerebral cortex in young, healthy individuals. Based on our findings, there is a need to study the effects of both ageing and brain diseases. Furthermore, since NIRS signals are affected by many physiological systemic functions, such as heart rate, respiratory rate, blood pressure, skin blood flow, motion artifacts [31,32], and bandpass filtering [33], correlation-based signal improvement [34] should be applied to separate components that are closely related to neural activity from NIRS data and should be measured extensively. In addition, since some tasks are affected by sex differences [35], it is necessary to study sex differences regarding the responsiveness of cerebral blood flow to oral stimulation in the future.

NIRS may detect emotional conditions in conjunction with other physiological biomarkers [36,37,38]. Furthermore, simultaneous measurement with NIRS and other neuroimaging techniques, such as fMRI [39,40] or EEG, may clarify whole-brain interactions. NIRS advancements in clinical settings have enabled the subjective assessment of not only discomfort and dysesthesia [41] but also comfort. Thus, the fit of dental prostheses, such as full-metal crowns and dentures, can be detected using the hemodynamics of the cerebral cortex as an indicator [42]. This study revealed that brain activity is closely related to oral stimulation, and that qualitative differences in oral stimuli result in different hemodynamic patterns in the cerebral cortex. Accordingly, NIRS-based recording of brain activity may serve as an objective method to evaluate oral function and contribute to the development of effective oral functioning training programs. New research and development initiatives in the field of dental medicine based on brain–oral function connections are anticipated.

## 5. Conclusions

This study demonstrated that tongue pressure against the anterior palatal mucosa elevates Oxy-Hb levels in the right somatosensory motor area and showed the deactivation of the PFC with passive, nontaste oral tactile stimulation. TPM probe insertion in the oral cavity activated the bilateral somatosensory motor area and deactivated the wide area of the APFC. Moreover, MTP_60%_ via the TPM balloon probe activated the bilateral somatosensory and motor cortex areas. These findings expand our understanding of brain hemodynamics during oral function. The development of a clinically feasible NIRS protocol during oral rehabilitation could facilitate the visualization and digitization of oral function from a cerebral perspective.

## Figures and Tables

**Figure 1 brainsci-12-00296-f001:**
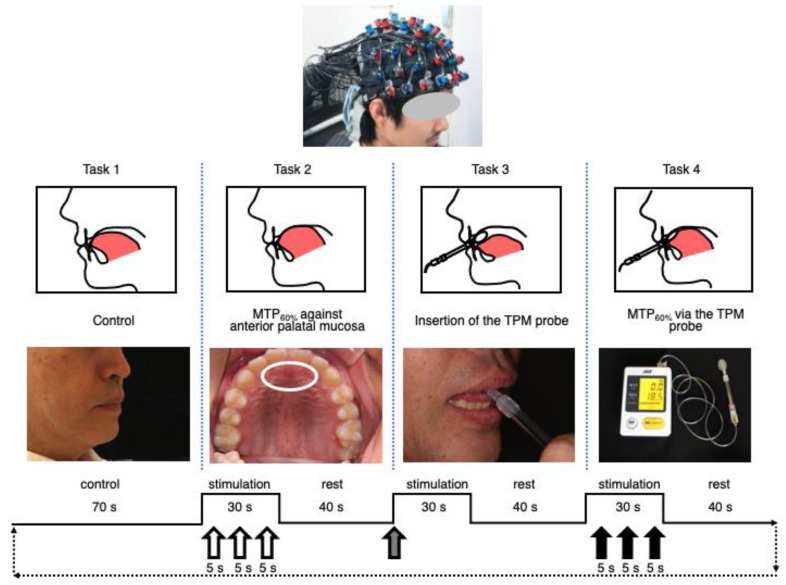
Task paradigm. The experimental design consisted of four 70 s task blocks. NIRS holder on the head during the four task measurements (top). Schematic depiction of the tongue posture during each task (second line). Images of the control stature, anterior palatal mucosa (white circle), TPM balloon-type probe, and TPM device (third line). The black line indicates sequential 30 s stimulation periods and 40 s rest periods (bottom). Task 1: resting tongue position for control. Task 2: MTP_60%_ against anterior palatal mucosa; 5 s, three times (open arrow). Task 3: insertion of the TPM probe (gray arrow). Task 4: MTP_60%_ via the TPM probe; 5 s, three times (solid arrow). NIRS, near-infrared spectroscopy; TPM, tongue pressure measurement, MTP, maximum tongue pressure; MTP_60%_, approximately 60% of the MTP for the experimental tongue pressure.

**Figure 2 brainsci-12-00296-f002:**
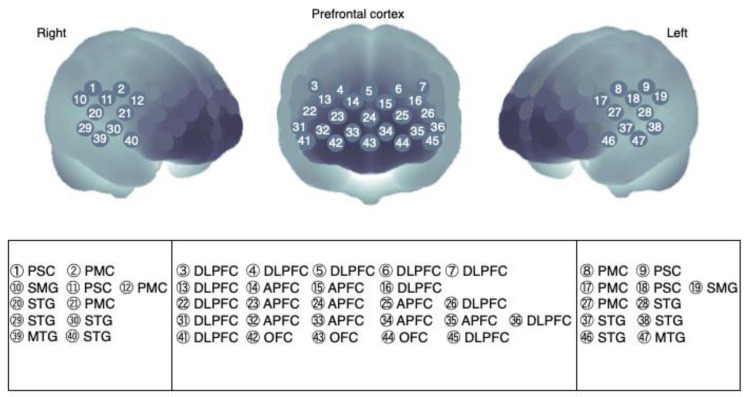
Channel allocation on the cerebral cortex. The encircled number represents near-infrared spectroscopy (NIRS: ETG-7100) 47 channels around the right temporal, frontal, and left temporal regions. PSC, primary sensory cortex; PMC, primary motor cortex; SMG, supramarginal gyrus; STG, superior temporal gyrus; MTG, middle temporal gyrus; DLPFC, dorsolateral prefrontal cortex; APFC, anterior prefrontal cortex; OFC, orbitofrontal cortex. Brain cortex schema was designed based on the report by Tsuzuki, et al. [23].

**Figure 3 brainsci-12-00296-f003:**
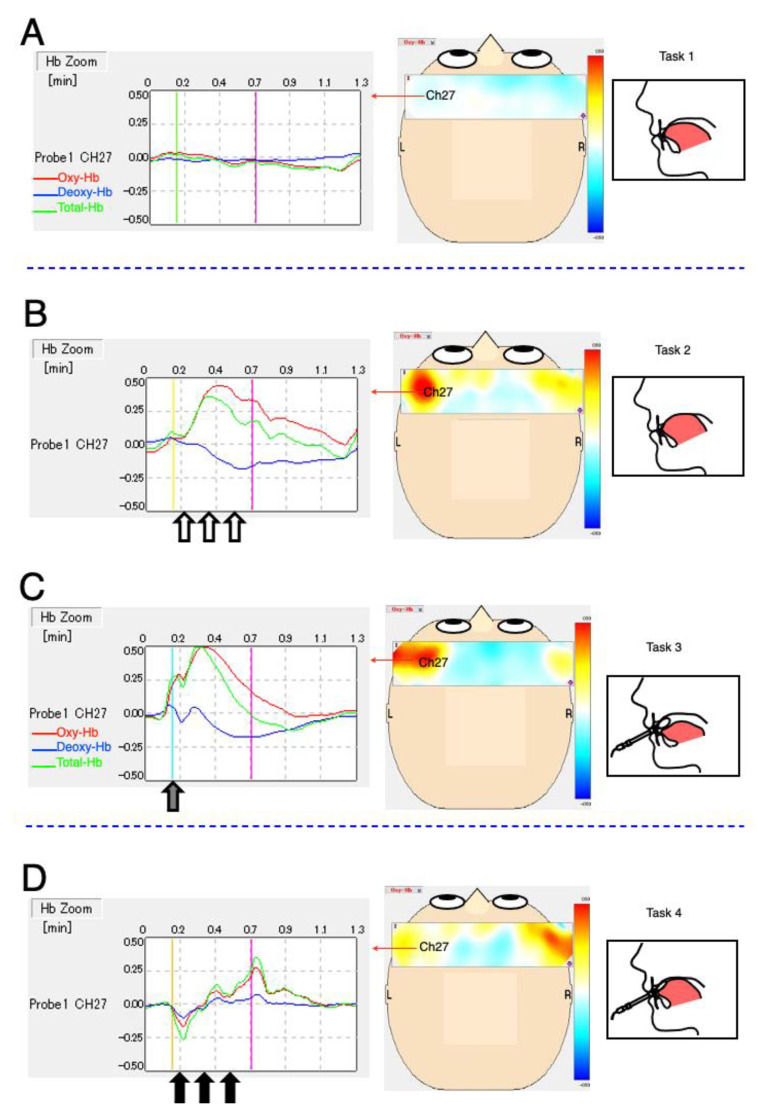
Typical waveform and topography. The left figures show near-infrared spectroscopy signal-integration waveforms at channel 27 around the left primary motor cortex; oxygenated hemoglobin (Oxy-Hb, red); deoxygenated hemoglobin (Deoxy-Hb, blue); and total hemoglobin (Total-Hb, green). Horizontal axis: time (min). The start of the tasks is indicated by four colors (light green, light yellow, turquoise-blue, and ocher, respectively), while the end of the tasks is indicated by pink. Vertical axis: Oxy-Hb values using arbitrary units (mM–mm). The center topography depict Oxy-Hb activation at the end of the tasks (pink) from overhead view and the range bar (±0.50 mm, mol–mm). (**A**) Resting tongue position for control. (**B**) MTP_60%_ against the anterior palatal mucosa, 5 s, three times (open arrow). (**C**) Insertion of the TPM probe (gray arrow). (**D**) MTP_60%_ via the TPM probe, 5 s, three times (solid arrow). MTP, maximum tongue pressure; MTP_60%_, approximately 60% of the MTP for the experimental tongue pressure; TPM, tongue pressure measurement.

**Figure 4 brainsci-12-00296-f004:**
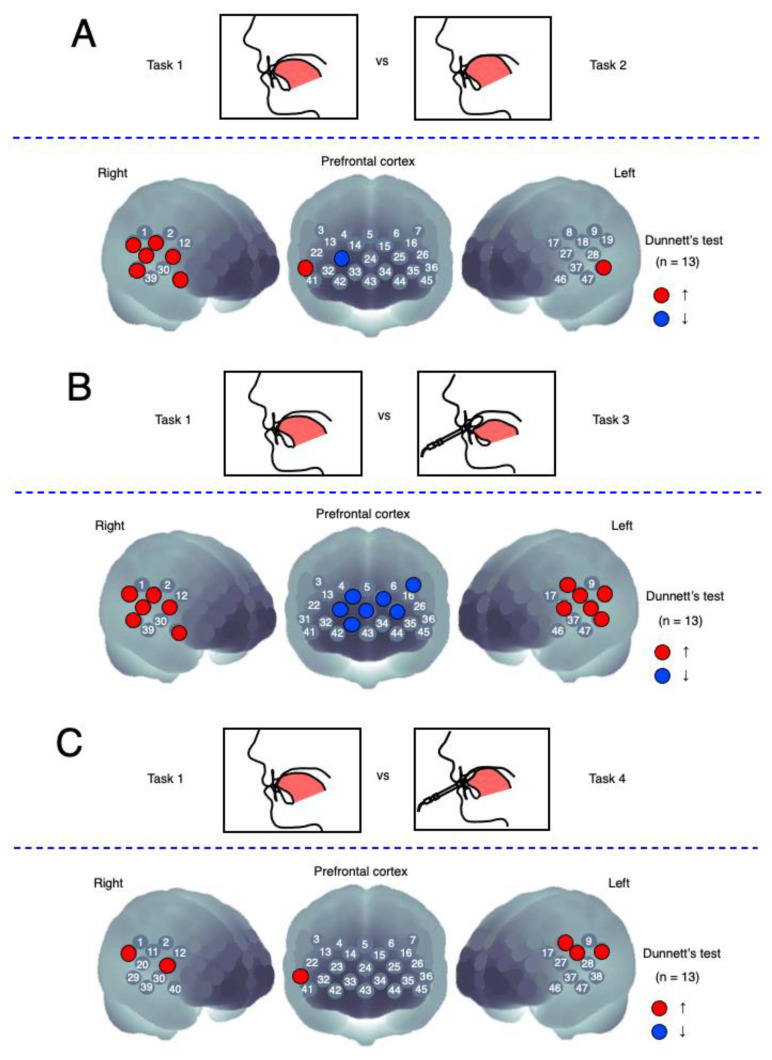
Activation analysis. Significant maps were represented by Dunnett’s post hoc test of oxygenated hemoglobin (Oxy-Hb) comparing Tasks 2, 3, and 4 with Task 1 (control task) (*n* = 13; Dunnett’s test after analysis of variance corrected false discovery rate, *p* < 0.05). Red-colored circles indicate significant increases compared with Task 1; blue-colored circles indicate significant decreases compared with Task 1. (**A**) Task 2 vs. Task 1; (**B**) Task 3 vs. Task 1; (**C**) Task 4 vs. Task 1.

**Figure 5 brainsci-12-00296-f005:**
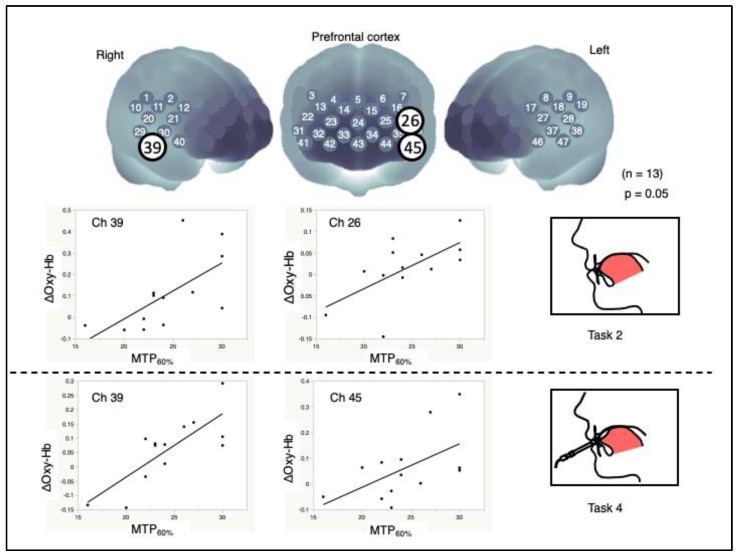
Correlation analysis. Functional connectivity between MTP_60%_ values and changes in oxygenated hemoglobin (ΔOxy-Hb). White-encircled numbers represent significant, positively correlated channels (*n* = 13; *p* < 0.05). Scatter plots of MTP_60%_ and ΔOxy-Hg deviation (bottom). MTP_60%_, approximately 60% of the MTP for the experimental tongue pressure.

**Table 1 brainsci-12-00296-t001:** Tongue pressure measurements.

Participants *n* = 13	Sex (kPa)	MTP1 (kPa)	MTP2 (kPa)	MTP3 (kPa)	Test Pressure (kPa)
a	M	43.4	48.7	56.5	30.0
b	M	50.4	49.6	51.0	30.0
c	M	41.5	41.5	41.6	24.0
d	M	40.8	37.3	41.9	24.0
e	M	20.5	16.7	27.8	16.0
f	M	32.4	39.1	37.0	20.0
g	F	40.2	43.4	40.3	22.0
h	F	35.4	41.7	41.0	23.0
i	M	52.2	53.7	50.8	30.0
j	M	41.4	43.7	43.8	26.0
k	M	35.4	35.2	36.7	22.0
l	M	36.1	36.5	38.9	23.0
m	F	47.7	48.1	45.6	27.0
		**Male**	**Female**	***p*-Value**	
**MTP (Mean ± SD, kPa)**		40.7 ± 9.0	42.6 ± 4.1	*p* = 0.74	

Participants performed the maximum tongue pressure (MTP) task three times. Experimental tongue pressures (MTP_60%_) were adjusted to approximately 60% of the mean MTP. kPa, kilopascals.

## Data Availability

Not applicable.

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
