# Peer review of "Effects of Tongue Pressure on Cerebral Blood Volume Dynamics: A Functional Near-Infrared Spectroscopy Study"

_brainsci, 2022, doi:10.3390/brainsci12020296_

Round 1
Reviewer 1 Report
This study seemed like a pilot study about the relationship between Oral functional training and cognitive functions.
I expect further study about whether of increasing brain metabolite in somatosensory and motor cortex of patients with Mild cognitive impairment by using Oral rehabilitation.
Author Response
Please see the attached response.

Reviewer 2 Report
The Authors present a study here that investigates the plausible relationships between oral tasks/stimuli and regional cortical activation. The research question is interesting and is related to a relevant and current field. The quality of writing of the manuscript is high, with high quality figures that provide great illustrations to the main text. With that in mind, unfortunately the study also suffers from some methodological and conceptual issues (see below) that I feel are need to be addressed before it can be accepted for publication.
- For me the most severe limitation of this study is that it is not established well enough that the recorded changes in cerebral hemodynamics are indeed a result of altered neural activity and not some other physiological function (i.e. skeletal muscle activity, skin blood flow, increased heart rate, etc.). My suspicions are further strengthened by the results shown on Figure 4 panel C - it appears that increased activation was found over bilateral temporal regions that also correspond to (in part) to the attachment of the temporal muscle. The observed results may arise from increased activity in the muscle and not neural activity; I am not saying that this is the case, only that it is not excluded or even considered in the study. It is also not in contradiction with the observed correlations, as increased muscle activity might also be associated with increased hemodynamic response. This issue is also related to a lacking (at least in description) of pre-processing steps applied to raw NIRS data, see my point below.
- From the manuscript it appears that the Authors did not apply any pre-processing steps to raw NIRS data. This is quite unfortunate, as NIRS signals are affected by many physiological systemic functions such as heart or respiratory rate, blood pressure, skin blood flow, motion artefacts and so on. These should be eliminated as they can act as confounders in the study (see e.g. Tachtsidis et al. 2008, Cooper et al. 2012). Band-pass filtering is commonly applied to remove periodic oscillations originating from heart rate and respiration (see e.g. Tian et al. 2009). Also, one might also apply correlation-based signal improvement (Cui et al. 2010) to separate a component of raw NIRS data that is more closely related to neural activity. Did the Authors apply any preprocessing to raw NIRS data? If yes then please include details in the manuscript, if not then I suggest to re-run the analyses on appropriately pre-processed data. I understand that it is beyond the limitations of the current dataset, but for future studies the Authors might want to consider simultaneous recording of systemic physiological functions such as heart rate (ECG), respiratory rate, continuous blood pressure or skeletal muscle activity (via electromyography) to be able to better establish a direct relationship between tongue/oral activity and regional cortical activity.
- The statistical analysis of the obtained data and results is lacking (at least in description). For example, did the Authors applied any adjustment (such as False Discovery Rate of Benjamini and Hochberg) for multiple comparisons? Were assumptions (normality, etc.) of the applied statistical tests confirmed? I also think that statistical analysis could be better designed. From the presentation of the results it appears that the study has one control condition ('Task 1') and three test conditions. These could be tested with a repeated measures ANOVA (if applicable considering the assumptions of ANOVA, given that all conditions came from the same subjects) to test for the main effect of state (Task), and then apply Dunnett's post hoc comparisons to compare all test conditions to the one control condition. Also, given that the same analysis (comparison) is performed for all channels, the Authors should apply adjustments during the statistical analyses, such as FDR or Bonferroni.
- I think the description of the experimental procedure could be improved. It is quite hard to understand the timing of the task - it is stated that the recording takes up to 19 minutes, however on the figures the pipeline shows that a recording session (Task1--->Task4) is 4x70=280s. It is also stated that the task block was repeated 4 times, which makes for 4x280=1120s, which is approximately 19 minutes, but then how results for the conditions were obtained exactly for each subject? All 4 blocks were used individually, or were blocks averaged per subject to obtain an average hemodynamic response? Please clarify this! Also, what exactly is Task 2? Pressure of 60% of MTP without the probe? If no probe is used here, then how it is ensured that the pressure is 60% of the maximal? And how exactly the 100% was measured? In a separate session before the 4 stimulation blocks? Please clarify!
- Also, for me it was hard to understand that Task 3 is not an active task, but it is a sensory stimulation. It was quite confusing especially considering Figure 3 panels B, C nd D, where hemodynamic responses of Task 2 and 3 (60% pressure and no pressure, just probe) are quite similar, while that of Task 4 (60% pressure similarly to Task 2, but with the probe inserted) are quite different. I would expect a similar response in Tasks 2 and 4. The Authors might improve the manuscript by better explaining the various 'Task' conditions, or use a different notation for Task 2 (e.g. sensory stimulation). Also I think it would be introductory to mention/better explain in the introduction that sensory stimulation also elicits increased neural activity/hemodynamic response in various brain regions, even though it might be obvious.
- It is mentioned multiple times in the introduction that investigating relationship between oral functions and cortical activity might be important in conditions related to aging, however the subject population of the study is consisted of young healthy participants. I think it would be instructive to flash this aspect out in the discussion, how the obtained results might be important related to healthy aging and what future plans the Authors have in that direction.
I hope these comments will help the Authors improve their manuscript. I think the manuscript carries significant values, but the above listed concerns must be addressed before the work can be accepted for publication.
Author Response
Please see the attached response.

Reviewer 3 Report
This is a very interesting study that examines the effect of tongue pressure on the cortical hemodynamics of human subjects using functional Near Infrared Spectroscopy. Changes in cortical hemodynamics in several areas such as prefrontal, temporal and motor regions based on different tasks were investigated. Correlations between cortical hemodynamics in different regions and MTP measurements were further shown. There are number of comments and concerns that need to be addressed in this manuscript:
- Number of subjects and gender distribution: The study is using only 13 subjects without a good distribution of male and female. Would that be possible to add additional subjects, especially females to this study?
- Statistical analysis:
- Authors used Paired t-test to compare task 1 results with respect to tasks 2, 3 and 4. Given that there are four number of measurements (Tasks) over 47 channels, the multivariate analysis that will account for between and within subject analysis would be more compelling. This would be especially useful to compare other tasks such as task 2 vs 3 and 2 vs 4 etc.
- Authors should further input gender as a factor and see if there are any differences between male and female with respect to both MTP and hemodynamic measurements. This is based on the number of studies showing the significant difference between male and female in tongue pressure that could possibly affect the cortical activation as well.
- Please add a separate section for statistical analysis and provide some more details such as what type of correlation analysis has been used (e.g. Pearson, Spearman, two tailed vs one tailed).
- Please further explain the source of similar activation pattern during task 2 and task 3, both showing left PFC activation.
- Please clarify if the average value over the 10 s pre-time, 35 s recovery and 4 second post time was used as a baseline for each task?
- Please elaborate on the difference between cortical activation (Left vs Right PFC) in Task 2 vs task 4. What authors hypothesize to be behind this result given that the main difference between these two tasks is the presence of TPM probe but the nature of the task is more or less the same
- This study is showing the effect of tongue pressure on cortical activation. It seems that tongue pressure can generally induce unwanted activation in fNIRS studies if the subjects unknowingly apply tongue pressure while performing the task. It would be great if authors can provide some insight regarding this issue in the discussion.
- There is a need for more elaborated discussion with respect to some of the results such as correlation analysis in the discussion section and other issues as stated above.
Author Response
Please see the attached response.

Round 2
Reviewer 2 Report
Thank you for addressing my points. I would like to point out some issues that I still have concerns with.
- Block averaging is indeed a common practice in NIRS-based studies, as it can help extracting the neural response. However, this is based on the fact (assumption) that the neural response is time-locked to the stimulus while background activity/noise is not, and thus by averaging the background activity will eventually cancel out given a sufficient number of stimulation blocks. However, this approach breaks down when the source of the noise is also time-locked to the stimulus, which can be the case here; if increasing tongue pressure is accompanied with flexion of other skeletal muscles, then the noise induced by this activation will not cancel out by averaging.
- I would also argue that even though performing the task does not include mandibular movement, it by no means miplies that muscles operating the mandibula remain idle. Quite the contrary, since tongue pressure is applied against the palatal mucosa, some muscles must be activated (to lesser or greater extent) in order to fix the position of the mandibula, given that several of the tongue's muscles are attached to the mandibula itself (e.g. genioglossus). Nevertheless, this muscle activity is very unlikely to be a confounder in task 3 (only probe, no pressure). I would still suggest the Authors at least discuss these issuee in the discussion/limitations.
- I don't understand why re-analysis of pre-processed data would require any re-experiments. Don't the Authors have the raw datasets recorded from the 13 subjects? Most general pre-processing steps applied to NIRS data are carried out offline, after the data is collected. Therefore, I see no reason why the Authors couldn't re-analyze their data after applying a band-pass filter (e.g. with cutoff frequencies 0.01-0.2Hz) to exclude confounding effects from heart pulsations and breathing cycles. The suggested CBSI procedure is not a necessity, I just suggested it as an example from the plethora of plausible pre-processing methods commonly applied to NIRS data. It also quite simple to implement, see Eq. (2) in Cui et al. 2010. Naturally, redoing of the recordings with simultaneous monitoring of other physiological measures such as ECG is out of the bounds of this study.
- Is there any reason why a strong lateralization was found in Task 2?
- I suggest including the reference Csipo, Tamas, et al. "Assessment of age-related decline of neurovascular coupling responses by functional near-infrared spectroscopy (fNIRS) in humans." Geroscience 41.5 (2019): 495-509. in the discussion when stated 'Since the response of cerebral blood flow may be altered by aging...'.
Reviewer 3 Report
I appreciate the authors' response and I found the response to be adequate.
